# AUGMENTATION - INTERPOLATIVE AUTOENCODERS FOR UNSUPERVISED FEW-SHOT IMAGE GENERATION

## ABSTRACT

We aim to build image generation models that generalize to new domains from few examples. To this end, we first investigate the generalization properties of classic image generators, and discover that autoencoders generalize extremely well to new domains, even when trained on highly constrained data. We leverage this insight to produce a robust, unsupervised few-shot image generation algorithm, and introduce a novel training procedure based on recovering an image from data augmentations. Our Augmentation-Interpolative AutoEncoders synthesize realistic images of novel objects from only a few reference images, and outperform both prior interpolative models and supervised few-shot image generators. Our procedure is simple and lightweight, generalizes broadly, and requires no category labels or other supervision during training.

## 1 INTRODUCTION

Modern generative models can synthesize high-quality (Karras et al., 2019; Razavi et al., 2019; Zhang et al., 2018a), diverse (Ghosh et al., 2018; Mao et al., 2019; Razavi et al., 2019), and high-resolution (Brock et al., 2018; Karras et al., 2017; 2019) images of any class, but only *given a large training dataset for these classes* (Creswell et al., 2017). This requirement of a large dataset is impractical in many scenarios. For example, an artist might want to use image generation to help create concept art of futuristic vehicles. Smartphone users may wish to animate a collection of selfies, or researchers training an image classifier might wish to generate augmented data for rare classes. These and other applications will require generative models capable of synthesizing images from a large, *ever-growing* set of object classes. We cannot rely on having hundreds of labeled images for all of them. Furthermore, most of them will likely be *unknown* at the time of training.

We therefore need generative models that can train on one set of image classes, and then generalize to a new class using only a small quantity of new images: *few-shot image generation*. Unfortunately, we find that the latest and greatest generative models *cannot even represent* novel classes in their latent space, let alone generate them on demand (Figure 1). Perhaps because of this generalization challenge, recent attempts at few-shot image generation rely on undesirable assumptions and compromises. They need impractically large *labeled* datasets of hundreds of classes (Edwards & Storkey, 2016), involve substantial computation at test time (Clouâtre & Demers, 2019), or are highly domain-specific, generalizing only across very similar classes (Jitkrittum et al., 2019).

In this paper, we introduce a strong, efficient, *unsupervised* baseline for few-shot image generation that avoids *all* the above compromises. We leverage the finding that although the latent spaces of powerful generative models, such as VAEs and GANs, do not generalize to new classes, the representations learned by autoencoders (AEs) generalize extremely well. The AEs can then be converted into generative models by training them to interpolate between seed images (Sainburg et al., 2018; Berthelot et al., 2018; Beckham et al., 2019). These Interpolative AutoEncoders (IntAEs) would seem a natural fit for few-shot image generation. Unfortunately, we also find that although IntAEs can reproduce images from novel classes, the ability to interpolate between them breaks down upon leaving the training domain. To remedy this, we introduce a new training method based on data augmentation, which produces smooth, meaningful interpolations in novel domains. We demonstrate on three different settings (handwritten characters, faces and general objects) that our *Augmentation-Interpolative Autoencoder* (AugIntAE) achieves *simple, robust, highly general, and completely unsupervised* few-shot image generation.

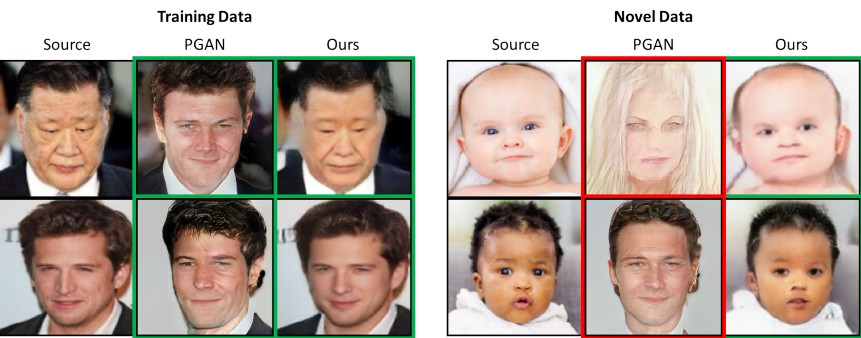

Figure 1: A basic image reconstruction task. One can either use a state-of-the-art PGAN (Karras et al., 2017), optimizing the latent code to match the generation to the input, or use our simpler AugIntAE. Both succeed on the training set of adult faces (left), but on the novel domain of baby faces (right), the best PGAN reconstruction ages the baby. Ours is far more faithful.

## 2 RELATED WORK

### 2.1 GENERATIVE MODELING

AEs were originally intended for learned non-linear data compression, which could then be used for downstream tasks; the generator network was discarded (Kramer, 1991; Hinton & Salakhutdinov, 2006; Masci et al., 2011). VAEs do the opposite: by training the latent space toward a prior distribution, the encoder network can be discarded at test time instead. New images are sampled directly from the prior (Kingma & Welling, 2013). Subsequent models discard the encoder network entirely. GANs sample from a noise distribution and learn to generate images which fool a concurrently-trained real/fake image discriminator (Goodfellow et al., 2014). Bojanowski et al. (2017) and Hoshen et al. (2019) treat latent codes as learnable parameters directly, and train separate sampling procedures for synthesizing the novel images.

Recent work has also seen a return to AEs as conditional generators, by training reconstruction networks to interpolate smoothly between pairs or sets of seed images. This is accomplished by combining the reconstruction loss on seed images with an adversarial loss on the seed and interpolated images. Different forms of adversarial loss (Sainburg et al., 2018; Berthelot et al., 2018) and interpolation (Beckham et al., 2019) have been proposed.

While all of these approaches generate new images, it is unclear if any of them can generalize to novel domains. Some results suggest the opposite: a VAE sufficiently powerful to model the training data becomes incapable of producing anything else (Bozkurt et al., 2018).

### 2.2 FEW-SHOT IMAGE GENERATION

Current attempts at few-shot image generation span a wide range of approaches and models. Neural Statistician, an early attempt, is similar to the AE in that it is built for few-shot classification, and largely discards the generative capability (Edwards & Storkey, 2016). Generation-oriented iterations exist, but likewise depend on a large, varied, labelled dataset for training (Hewitt et al., 2018). Other approaches based on few-shot classification include generative matching networks (Bartunov & Vetrov, 2018) and adversarial meta-learning (Clouâtre & Demers, 2019). These models also depend on heavy supervision, and are fairly complicated, involving multiple networks and training procedures working in tandem - making them potentially difficult to train in practice reliably.

Separate work has approached few-shot image generation from the side of generative modeling. Wang et al. (2018), Noguchi & Harada (2019) and Wu et al. (2018) investigate the ability of GANs to handle domain adaptation via fine-tuning - thus requiring substantial computation, and more novel class examples than are available in the few-shot setting. Zhao et al. (2020) train GANs directly from few examples, though still more than are at hand for few-shot learning, and can be considered orthogonal work, as AugIntAE can serve as a useful pre-trained initialization. Antoniou et al. (2017) and Liu et al. (2019) use adversarial training to produce feed-forward few-shot generators. However,

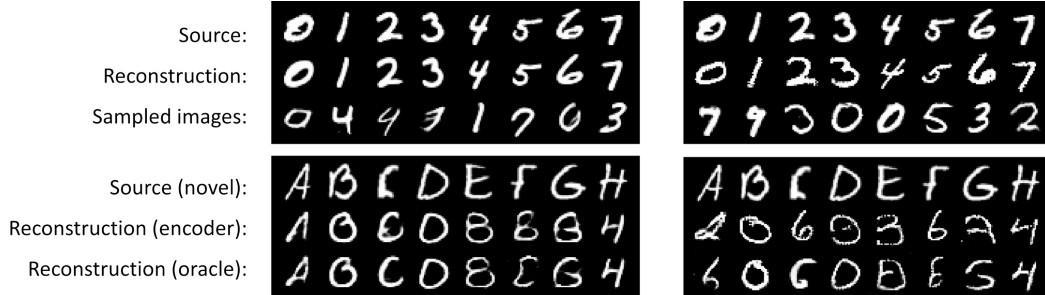

Figure 2: A VAE (left) and WGAN-GP (right) trained on MNIST (top row) successfully recover training images (row 2) and synthesize new ones (row 3). They struggle to represent images from novel classes (EMNIST letters, rows 4 and 5), even when latent codes are optimized directly, as an oracle encoder (row 6). VAE reconstructions use the mean embedding $\mu$ directly and do not sample in latent space. Best viewed digitally.

both models still depend on varied, labelled training data, and risk exhibiting the same problems as standard GANs: mode collapse and hyperparameter sensitivity (Arora et al., 2017; 2018).

Jitkrittum et al. (2019) introduce an algorithm for class-conditioning an unconditioned generative model. New images are produced by matching latent space batch statistics to real images from a single, possibly novel class. Nguyen et al. (2017) learn individual latent codes from a pretrained discriminator, while Wang et al. (2020) train a latent sampling network. These approaches have little to no evaluation on novel classes, and to what degree they generalize depends entirely on the pretrained image generator. They may also require substantial test-time computation. In contrast, AugIntAEs are lightweight, train robustly, and generalize broadly from completely unlabelled data.

## 3 PROBLEM SETUP

Let $X$ be a large, *unlabelled* collection of images depicting objects from a set of classes $C$. Let $X'$ be a very small set of images - as few as two - belonging to a novel class $c' \notin C$. Our goal is to train a network on $X$ which, given $X'$, generates images clearly belonging to $c'$. We refer to this as the network's ability to *generalize* to new domains (note that this usage is distinct from "generalizing" to novel images in the same domain, a much simpler task). We cannot directly adapt the network to $X'$ using SGD, as $X'$ contains too few images to prevent overfitting.

This is an extremely difficult problem, since it is unclear if a neural network trained to model the data distribution in $X$ can even *represent* images from a different distribution, let alone sample from it. Therefore, we first examine whether existing generative models can faithfully encode novel class images in latent space. We train a VAE and a WGAN-GP (Gulrajani et al., 2017) on MNIST handwritten digits (LeCun et al., 1998), as well as an encoder network that inverts the WGAN-GP as in Bau et al. (2019) (details in appendix). We then evaluate the ability of each generative model to recover particular images. Using the built-in VAE encoder and the WGAN-GP inversion network, we find that while both models can reconstruct training images (Fig. 2, top), the same approach fails on images from novel classes - in this case, EMNIST handwritten letters (Cohen et al., 2017). The outputs do not much resemble the inputs; crucial semantic information is lost (Fig. 2, bottom). To discount the possibility of sub-optimal encoders, we simulate an oracle encoder, refining the latent code parameters for each image directly via SGD. These reconstructions are not much better. Fig. 1 demonstrates a similar failure in a large, state-of-the-art pretrained GAN. This confirms prior findings (Bozkurt et al., 2018) that current generative approaches by default *cannot even represent* images from novel classes. Generating *new* novel class images is simply out of the question.

Why do sophisticated generative models fail to generalize? We argue this is largely by design. Generative models such as VAEs and GANs are trained to minimize the divergence between a prior distribution and a learned posterior distribution, where one or both are approximated by repeated sampling. VAEs push the learned latent posterior toward a Gaussian prior, while GANs map samples from the prior to a posterior distribution in image space. In both cases, latent vectors are repeatedly

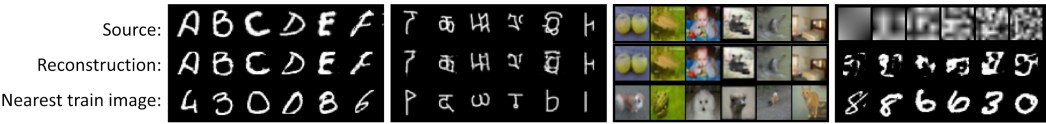

Source:
Reconstruction:
Nearest train image:

Figure 3: Randomly chosen AE image reconstructions for unseen classes. From left to right: MNIST generalizes to EMNIST, Omniglot train generalizes to Omniglot test, and CIFAR-10 generalizes to CIFAR-100, but MNIST does not generalize trivially to white noise. Bottom row: nearest (in pixel space) training image is shown for comparison. Best viewed digitally.

Table 1: Average AE per-pixel L2 reconstruction error (normalized to [0, 255]). Rows give the training dataset, columns give the evaluation dataset, and cells measure generalization from row to column. Generally, error is low. The bottom row contains scores for a VAE, which generalizes poorly from MNIST. Implementation details in appendix.

| | MNIST | EMNIST | Omni (train) | Omni (test) |
|---|---|---|---|---|
| *# images:* | *60,000* | *124,800* | *19,280* | *13,180* |
| MNIST | 2.69 | 13.48 | 9.83 | 11.58 |
| EMNIST | 6.59 | 3.89 | 8.31 | 9.79 |
| Omniglot (train) | 14.74 | 21.47 | 2.13 | 8.30 |
| Omniglot (test) | 15.01 | 21.44 | 8.04 | 2.32 |
| MNIST (VAE) | 5.92 | 23.10 | 15.98 | 17.91 |

| | CIFAR-10 | CIFAR-100 |
|---|---|---|
| *# images:* | *50,000* | *50,000* |
| CIFAR-10 | 7.58 | 9.09 |
| CIFAR-100 | 9.04 | 7.98 |

sampled and sent through the generator. Thus, by the time the generator is trained to convergence, and the posterior approaches the prior (or vice-versa), every region of the latent space feasible under the prior will have been mapped at some point to a training image - or, in the case of GANs, to an image indistinguishable from a training image. This means that a properly trained VAE or GAN cannot construct or reconstruct new object classes. If it could, then it would have been able to sample such images during training - which would mean it had not been properly trained at all.

## 4    AUGMENTATION-INTERPOLATIVE AUTOENCODERS

**AutoEncoders**: As discussed above, minimizing the divergence between prior and posterior training distributions ensures good image synthesis, but poor generalization. The opposite could also hold: AEs do not enforce any latent distribution on the data posterior, and so might generalize well. More formally, given a network $E$ that maps an image $x$ to latent vector $z$, and a generator $G$ mapping $z$ back to image space, we refer to the function composition $G(E(\cdot))$ as the autoencoder. $E$ and $G$ are trained jointly over $X$ to minimize the pixel reconstruction error between $x \in X$ and $G(E(x))$. The question of generalization becomes, to what degree does a trained AE maintain close proximity between $x'$ and $G(E(x'))$ for $x'$ which lies far from the data manifold of $X$?

By this measure, we find that our conjecture holds: AEs generalize surprisingly well. Examples are given in Fig. 3, demonstrating near-perfect generalization performance over three pairs of class-disjoint datasets: MNIST digits to EMNIST letters, Omniglot training alphabets to Omniglot testing alphabets (Lake et al., 2015), and CIFAR-10 to CIFAR-100 (Krizhevsky et al., 2009). We also quantitatively evaluate generalization (in terms of reconstruction error) between all the above pairs, as well as between Omniglot and MNIST/EMNIST, which includes domain shifts, e.g., stroke width (Table 1). Reconstruction quality is high across the board, especially given that the MNIST and CIFAR-10 networks learn only ten distinct classes! AEs exhibit very little overfitting to the training domain, learning a general mapping despite heavy class constraints.

It is possible that this generalization is a result of AEs simply learning an identity function. Fortunately, this is not the case: AEs learn clear image priors. We find that our trained AEs are much more effective at encoding real images than noise (see Fig. 3, right). We also find that low-frequency noise is encoded more faithfully than high-frequency noise - an analysis is provided in appendix. The learned AE mapping, while general, is also nontrivial.

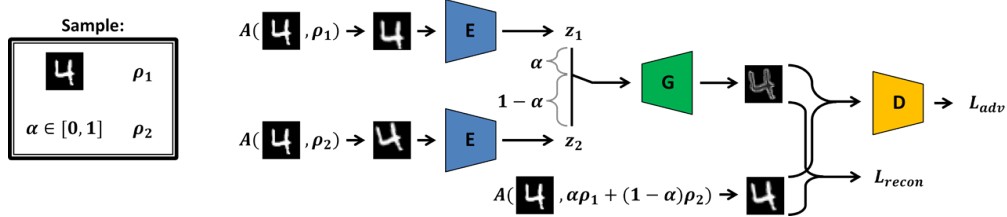

Figure 4: Training for AugIntAE. For every training image, two sets of augmentation parameters $\rho$ (rotation, translation, etc.) are sampled, plus a mixing coefficient $\alpha$. The network must recover the mixed transformation image from the same mixture in latent space. An auxiliary adversarial discriminator improves quality.

**Interpolative AutoEncoders**: The fact that AEs generalize suggests they are capable of acting as few-shot image generators for novel classes, given a method for sampling appropriate latent $z$ vectors. One possibility is to interpolate between data points in latent space: every sampled point is a weighted sum of two or more real points. This allows us to produce novel combinations of seed images without changing the semantic content. Unfortunately, it is a known fact that AEs do not interpolate well, as shown in Fig. 5, row 2 (Berthelot et al., 2018). Prior work (Sainburg et al., 2018; Berthelot et al., 2018; Beckham et al., 2019) addresses this by applying an adversarial loss to the interpolated images, which works well in the training domain. However, we find the Interpolative AutoEncoder (IntAE) approach overly restrictive for our purposes: it constrains all interpolations between arbitrary image pairs to the training domain. For example, on MNIST, an IntAE must produce recognizable digits when interpolating between a 3 and a 4, a semantically unintuitive result. This makes the learning process harder and causes the model to learn interpolations that do not generalize. When it interpolates between letters (Fig. 5, row 3, left), we find that it *does* produce undesirable artifacts - that look like numbers!

**Augmentation-Interpolative AutoEncoders**: We introduce a novel training procedure for IntAEs to remove such artifacts while maintaining generalizability. Instead of optimizing interpolated images using only a GAN loss, we train the network to directly recover known, semantically interpolated images from the corresponding interpolated latent code. This accomplishes two things simultaneously: first, we only interpolate between images where the interpolation makes semantic sense, since we must know the interpolated image in advance. This simplifies the learning problem significantly. Second, the model is no longer constrained to the training manifold when interpolating arbitrary training image pairs. The network can now learn simpler, more direct interpolations that work well on both training and novel domains.

Formally, suppose we have a triplet of images $A = f(\rho_1)$, $B = f(\rho_2)$ and $C = f(\alpha\rho_1 + (1-\alpha)\rho_2)$, where $f$ is some unknown image generating process, $\rho$ is a semantic variable, and $\alpha \in [0, 1]$. Using this triplet, we train the interpolative AE to reconstruct $C$ by decoding the interpolated latent code of $A$ and $B$. Formally, we train the encoder $E$ and the generator $G$ by minimizing:

$$L_{recon} = ||C - G(\alpha E(A) + (1-\alpha)E(B))||_1 \qquad (1)$$

In practice, finding appropriate image triplets $A, B, C$ in a dataset of independent images is difficult. Instead, we *synthesize* $A, B, C$ using affine spatial transformations, and color jitter for 3-channel images. Given training image $x$, we randomly sample two sets of augmentation parameters (translation, rotation, hue, etc.). Applying each of these transformations independently to $x$ yields $A$ and $B$ (for example, a $10^o$ rotation and a $-5^o$ rotation). We then sample a weight $\alpha \in [0, 1]$ and compute a weighted average of the two transformations, which we apply to $x$ to produce $C$ (in our example, if $\alpha = \frac{1}{3}$, $C$ represents a $5^o$ rotation). The Augmentation-Interpolative AutoEncoder (AugIntAE) is then trained to recover $C$ from the $\alpha$-weighted interpolation between the latent embeddings for $A$ and $B$. This corresponds to Equation 1.

We can also augment the model with the original IntAE losses: a reconstruction loss on $A$ and $B$, and a GAN loss $L_{adv}$ on the interpolated $C$. In practice, we found that the former did not noticeably affect performance, while the latter was helpful in reducing the blurriness of output images. Subsequent models include $L_{adv}$. The full procedure is displayed in Fig. 4.

Table 2: FID score and train/test classification error across models and datasets. Misclassification rate is measured in percentage points and captures likelihood of belonging to the wrong domain (training data). AugIntAE is superior in all contexts. GANs trained on train/test images and evaluated on test images are given as baselines/oracles.

| **FID** ($\downarrow$) | WGAN-GP (train) | AE | IntAE | AugIntAE | WGAN-GP (test) |
|---|---|---|---|---|---|
| MNIST $\rightarrow$ EMNIST | 25.0 | 16.7 | 21.5 | **12.3** | 8.3 |
| Omniglot (train $\rightarrow$ test) | 86.9 | 87.1 | 86.7 | **82.5** | 80.0 |
| CelebA (male $\rightarrow$ female) | 86.9 | 26.1 | 24.0 | **21.1** | 15.4 |
| CIFAR-10 $\rightarrow$ CIFAR-100 | 54.3 | 53.8 | 52.1 | **47.9** | 42.0 |
| **Misclassification rate** ($\downarrow$) | WGAN-GP (train) | AE | IntAE | AugIntAE | WGAN-GP (test) |
| MNIST $\rightarrow$ EMNIST | 96.5 | 4.1 | 2.7 | **2.4** | 0.5 |
| Omniglot (train $\rightarrow$ test) | 78.5 | 9.4 | 10.3 | **7.9** | 24.8 |
| CelebA (male $\rightarrow$ female) | 89.6 | 10.6 | 29.3 | **8.9** | 8.3 |
| CIFAR-10 $\rightarrow$ CIFAR-100 | 72.9 | 18.5 | 19.8 | **13.7** | 15.3 |

At first glance, learning the space of affine and color transformations does not appear particularly helpful for an IntAE. Very few visual relationships in the real world can be captured by these transformations alone. However, we find that these learned interpolations act as a powerful *regularizer* on the latent space, allowing AugIntAE to smoothly capture far more interesting and difficult transformations as well, such as shape, lighting, and even 3D pose.

**Few-shot generation**: Once the AugIntAE is trained, we can sample novel images given only a set of seeds. Simply select a random pair of images, find their latent space embeddings, sample $\alpha \in [0, 1]$, and generate the image from the $\alpha$-weighted mixture of embeddings. More sophisticated sampling techniques are possible, but left to future work.

## 5 EXPERIMENTS

All encoder/generator networks use standard 4- to 6-layer GAN architectures. We employ shallow networks to illustrate that AugIntAE itself, not network power, is responsible for good performance.

We use two baseline models. The first is an AE trained without interpolation but with the auxiliary GAN loss. We use an LSGAN (Mao et al., 2017) for the discriminator. The second baseline is an IntAE representing prior work: seed images are reconstructed while interpolations are trained via GAN. The GAN loss is also applied to the seed image reconstructions. The choice of LSGAN discriminator means that IntAE captures two prior models: it is an LSGAN instantiation of Beckham et al. (2019), and also a version of Berthelot et al. (2018) with discretized labels. We use the same data augmentation in all models and set parameters as large as possible without introducing significant visual artifacts. Training and evaluation details for all experiments are in the appendix.

### 5.1 QUANTITATIVE RESULTS

We report quantitative scores for all results in Table 2. We examine four train/test dataset pairs: two handwritten character settings and two natural image settings. For each pair we report two metrics: FID score, which captures overall image quality, and test set classification rate, which captures the degree of generalization and semantic faithfulness to the target domain. For the latter metric, we train a separate classifier on the combined train and test datasets to distinguish training images from testing images. Generators that generalize well to the test domain should have higher rates of test-set classification, while those that do not generalize will produce images skewed toward the training domain. On both metrics we find that AugIntAE is superior to AE and IntAE baselines in all settings. We also include standard GAN models trained solely on the training/testing datasets as baseline/oracle models, respectively. Our FID scores generally approach but do not exceed the oracle scores. We now examine individual dataset results in detail.

Figure 5: Examples of interpolation between handwritten character pairs from novel classes (EM-NIST left, Omniglot test right). AugIntAE interpolates smoothly, preserves the desired shape, and unlike the AE/IntAE baselines, produces minimal artifacts even when transformations are non-affine (far left). Best viewed digitally.

Figure 6: Comparison to few-shot generators. Each row corresponds to a distinct set of 5 seed images (left). Neural Statistician and DAGAN overfit to the training data. The AE successfully generates letters but introduces visual artifacts. AugIntAE more effectively removes these artifacts than the baseline IntAE. Results from AE-based models are midpoint interpolations between seed pairs. Best viewed digitally.

## 5.2 HANDWRITTEN CHARACTERS

**Image quality:** We evaluate the performance of AugIntAE on two handwritten character dataset pairs. One set of models trains on MNIST digits and evaluates on EMNIST letters, while the other transfers from the Omniglot training alphabets to the testing alphabets. The autoencoders in both cases use the 4-layer encoder/generator of InfoGAN (Chen et al., 2016), with latent dimensionality reduced to 32, in keeping with prior IntAE work (Beckham et al., 2019; Berthelot et al., 2018).

Fig. 5 shows example interpolations for these two contexts. AugIntAE produces better interpolations and removes visual artifacts, particularly discontinuities, present in the AE and IntAE images. These qualitative results are verified quantitatively in Table 2. AugIntAE, as measured by FID score (Heusel et al., 2017), outperforms all baselines. We also measure the semantic faithfulness of the interpolations to the test seed images, by training a separate classifier to distinguish training data from testing data. AugIntAE images are classified correctly more often than any baseline. These results hold not just for handwritten characters, but across *all* our testing regimes. In terms of both image quality and semantic fidelity, AugIntAEs are effective few-shot generators.

**Generalizability:** We compare AugIntAE as a few-shot generator to two additional baselines: Neural Statistician (Edwards & Storkey, 2016) and DAGAN (Antoniou et al., 2017). Both approaches require class labels on the training data, while ours does not. We train both models on MNIST and then attempt to synthesize new images from EMNIST. Fig. 6 makes clear that AugIntAEs generalize more broadly and are much less restricted by the narrow training domain. Neural Statistician and DAGAN overfit to the training classes, and generate number images instead of the desired letters.

**Data hallucination:** As a practical use case, we utilize AugIntAE for data augmentation. We train a series of classifiers on the letters in the "ByMerge" split of EMNIST (lower- and upper-case separated), each using a different interpolation strategy as data augmentation. Half of each training batch is augmented data, produced by interpolation between same-class images with the labels preserved and $\alpha = .5$. The model without augmentation use the same batch size but train for twice as long. All autoencoders used for interpolation are trained on MNIST. All augmentations provide gains, as shown in Table Table 3, including pixel-level interpolation, a constrained form of MixUp (Zhang et al., 2018b). However, the largest gain is obtained by interpolating in the latent space of an AugIntAE.

Table 3: Interpolation as a form of data augmentation on EMNIST. Results are averaged over ten runs, with 95% confidence intervals. AugIntAE provides the most effective augmentation.

| Augmentation | None | Mixup | AE | AugIntAE |
|---|---|---|---|---|
| Accuracy | $90.33 \pm .06$ | $90.66 \pm .04$ | $91.27 \pm .04$ | $\mathbf{91.50 \pm .05}$ |

Figure 7: Ablation study on forms of data augmentation. Each row corresponds to a distinct set of 5 seed images (left); results are midpoint interpolations between seed pairs. Each individual form of augmentation improves over the AE baseline (fewer artifacts), but none approach the performance of using them all together (right). See, for example, the "S" in the top-right corner of each plot. Best viewed digitally.

**Ablation:** Affine augmentation encompasses a range of independent transformations (rotation, translation, scale, skew) and so it is worth examining to what degree each is necessary. We train four MNIST AugIntAE models using each independent transformation as the sole augmentation technique. Sample outputs are given in Fig. 7, along with AE and full AugIntAE baselines for comparison. We find that interpolative training using each form of augmentation improves over the AE baseline, but at the same time no individual augmentation approaches the performance of the full AugIntAE: the different augmentations act synergistically.

## 5.3    CELEB-A

We extend our results to the more difficult domain of higher-resolution natural images. Our models train on the male faces from Celeb-A, scaled to $128 \times 128$ resolution, and generalize to female faces. Network architectures follow DCGAN (Radford et al., 2016) with an additional half-width layer (at the bottom/top of the encoder/generator, respectively) to handle the increased resolution. Latent dimensionality is 512, in keeping with prior IntAEs (Sainburg et al., 2018; Beckham et al., 2019).

The results, displayed in Fig. 8 and Table 2, are similar to our findings for handwritten characters. AEs generalize well, but produce visual artifacts during interpolation: unrealistic, transparent regions around the head. IntAE produces high-saturation color artifacts. Compared to these baselines, AugIntAE removes the artifacts and restores semantic meaning to the interpolation path, *even when the interpolation is non-affine* (as in Fig. 8).

One might wonder if interpolative sampling acts as a constraint on generated image variety. Note that the number of unique image pairs grows quadratically with the number of seed images, so that

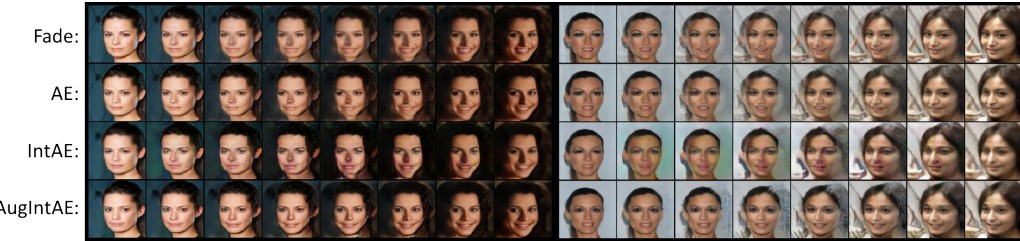

Figure 8: CelebA interpolations involving simultaneous pose and lighting shifts. AE and IntAE darken the nose unrealistically (left) or produce transparent artifacts (right), while AugIntAE accurately navigates the lighting/pose shift. Best viewed digitally.

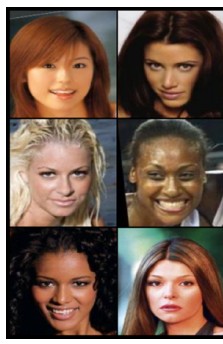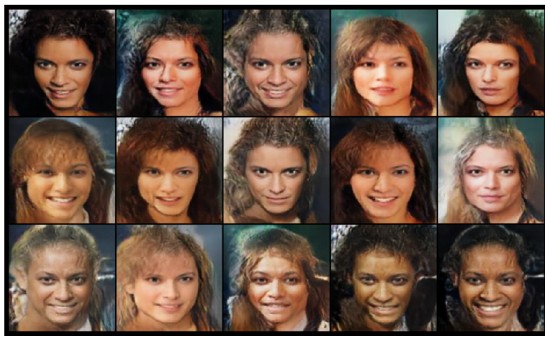

Figure 9: Six seed images (left) are sufficient to produce a variety of novel images (right). The fifteen displayed here represent the midpoints of the fifteen unique pairwise interpolation paths. We re-emphasize that this network was trained only on male faces. Best viewed digitally.

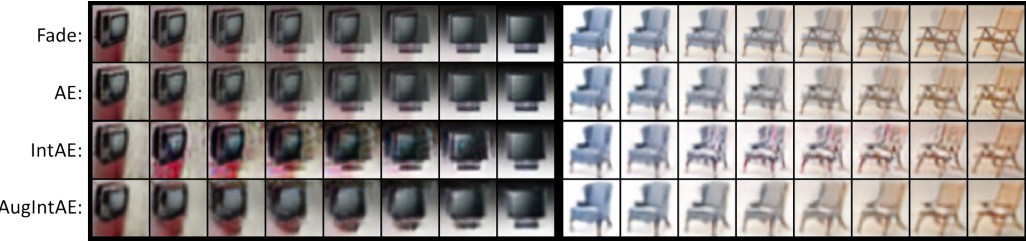

Figure 10: Examples of interpolation between image pairs from novel classes (CIFAR-100). AugIntAE interpolates smoothly with minimal transparency and color artifacts. Examples are hand-picked so that interpolation is possible. Best viewed digitally.

even with just six seeds AugIntAE can produce broad variety (Fig. 9). We conclude that AugIntAE is effective on higher-resolution color images, not just handwritten characters.

## 5.4 CIFAR

Finally, we evaluate our model in an extremely challenging setting: unconstrained natural images, and classes with large intra-class variation. The latter property is especially challenging for interpolation based models (e.g., how to interpolate between a real bear and a teddy bear?). We train our models on CIFAR-10 and evaluate on novel CIFAR-100 classes. Network architecture uses DC-GAN with 512 latent dimensions. Fig. 10 displays example outputs from AE, IntAE, and AugIntAE on novel CIFAR-100 data, and quantitative results are in Table 2. We conclude that AugIntAE is the most effective interpolative few-shot generator for natural images, though much work remains for the more difficult cases (see supplementary).

## 6 CONCLUSION

We introduce a powerful, lightweight, and label-free method for few-shot image generation. Building on the generalizability of AEs, we introduce a novel training procedure that leads to higher quality interpolations. The resulting AugIntAEs are robust, generalize far more broadly than prior few-shot generators, and are practically useful for downstream tasks.

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

# A APPENDIX

## A.1 ARCHITECTURES AND TRAINING

**Architectures**: Nearly all network backbones follow either InfoGAN, if operating on handwritten characters, or DCGAN, if operating on natural images. Celeb-A networks include an additional half-width layer at the bottom of the encoder, and at the top of the generator, to account for increased image resolution. For convenience, we use the same network architecture for both AugIntAE encoders and auxiliary GAN discriminators. The only difference is that the discriminator has one output neuron in the final layer, while encoders have as many as there are latent dimensions.

We $L_2$-normalize our latent features and perform interpolation via spherical linear interpolation (SLERP). Latent dimensionality is set to 32 for handwritten character models, which, given their $28 \times 28$ resolution, amounts to an almost 25-fold reduction in dimensionality. Celeb-A dimensionality is set to 512, with a $128 \times 128$ resolution and three color channels, producing a reduction in dimensionality of almost two orders of magnitude. CIFAR images use the standard DCGAN resolution of $64 \times 64$ and 512 latent dimensions for an almost 25-fold dimensionality reduction, similar to the handwritten character models.

VAEs, WGAN-GPs, and classifiers all use the same architectures, plus necessary modifications to the number of input/output neurons. WGAN-GPs draw from the same latent dimensionality as the corresponding AE models, and receive $L_2$-normalized noise samples rather than samples from a standard Gaussian. VAE encoders have output dimensionality twice the above, as they predict both a mean $\mu$ and variance $\gamma$ for each coordinate. Classifiers have as many output neurons as there are classes in the given task (2 for train/test dataset classifiers, 37 for EMNIST letter classifiers) and end in a softmax layer.

**Training**: Unless stated otherwise, all networks trained on real images use the Adam optimizer with initial step size .001. Models train for 100 passes over the given dataset. Models with no adversarial component cut the learning rate by a factor of 10 at epochs 35 and 70.

We chose these hyperparameters without a validation set, using convergence on the training data as our sole criterion.

**GAN training**: To stabilize learning, adversarial models do not change their learning rate. We also rescale all adversarial gradients so that their magnitudes match those of the reconstruction loss gradient. WGAN-GPs train for twice as long as other models, but update the generator only once per five discriminator updates.

On some datasets, we found that the auxiliary LSGAN discriminator could collapse and allow the generator to "win" the adversarial game. To prevent this, we introduce a scaling factor $k \in [0, 1]$ that updates dynamically based on discriminator performance. Specifically:

$$L_{AE} = L_{recon} + k * \gamma * L_{adv} \tag{2}$$

where

$$\gamma = \frac{\|\nabla L_{recon}\|_2}{\|\nabla L_{adv}\|_2} \tag{3}$$

calculated per-image, and $k$ is adjusted with each gradient update according to the following rules:

$$k_0 = 1 \tag{4}$$
$$\bar{k} = k_t - .001 * (1 - (D(x_{real}) - D(x_{fake}))) \tag{5}$$
$$k_{t+1} = max(0, min(1, \bar{k})) \tag{6}$$

This update scheme for $k$ ensures that whenever the scores coming from the discriminator $D$ for real and fake images are separated by less than 1 on average, $k$ decreases. The generator then focuses more on the reconstruction task and becomes less competitive, allowing the discriminator network

| Input $3 \times 512 \times 512$ RGB image |
|---|
| $4 \times 4$ conv. 32 BN ELU stride 2 |
| $4 \times 4$ conv. 64 BN ELU stride 2 |
| $4 \times 4$ conv. 128 BN ELU stride 2 |
| $4 \times 4$ conv. 256 BN ELU stride 2 |
| $4 \times 4$ conv. 512 BN ELU stride 2 |
| $4 \times 4$ conv. 1024 BN ELU stride 2 |
| FC 512 |

Table 4: Inversion network architecture for PGAN.

to "catch up" until the margin of separation is greater than 1 again. At that point $k$ increases back to 1, at which point the reconstruction and adversarial losses are equally weighted once more.

**Augmentation Parameters**: We sample data augmentation parameters $\rho$ uniformly over predefined ranges of values. For handwritten character datasets, we sample rotations in the range $[-20, 20]$, translations in the range $[-4, 4]$, scaling in the range $[.8, 1.2]$, and shear in the range $[-6, 6]$. These values were chosen heuristically: we picked the largest values that would not occlude or obscure the character.

Natural image AugIntAEs sample from half the range of rotation and skew, and double the range of translation (though because of the higher resolution, this comes out to much smaller displacement overall). These values were chosen so as not to introduce large regions of empty space into the image, while staying as large as possible. CIFAR images are symmetric-padded; Celeb-A images are center cropped and do not require padding. Additionally, we sample a range of color jitter parameters for AugIntAEs handling 3-channel RGB images. We sample brightness/contrast/saturation adjustments from the range $[.75, 1.25]$ and hue from the range $[-5\%, +5\%]$. We again chose these values to be as large as possible without producing unrealistic or unparsable images.

Similar to Sainburg et al. (2018), we found that the network learned interpolations near the seed images more easily than near the midpoint of the interpolation path. We therefore biased our sampled $\alpha$ toward the midpoint by sampling the mean of two uniform random draws.

To ensure that our baselines are trained on the same data distribution as our AugIntAEs, we use the $\alpha$-weighted interpolated image as the training image, even when the network does not use interpolative training. This only applies to AE and IntAE models.

### A.2 GAN Inversion

Obtaining generalization results for a GAN involves inverting the generator, which we attempt via a combination of a learned inversion network and direct SGD optimization on the latent codes for each target image. For the PGAN (fig. 1), we use a publicly available pretrained model provided by Facebook Research[1]. For the MNIST/EMNIST generalization experiments (fig. 2) we use our own implementations, constructed and trained using the procedure described above. The learned inversion network for the PGAN generator uses the architecture described in Table 4, while the inversion network for the MNIST WGAN-GP uses the same encoder as in other experiments. Both networks are trained by sampling images from the generator, and attempt to reconstruct the corresponding latent code for each image using mean squared error. We use SGD with an initial step size of .0001 and momentum equal to .1. We train for 6400 iterations and cut the learning rate by a factor of 10 after every 1280 iterations.

The subsequent stage of inversion involves direct refinement of the latent codes provided by the inversion network via SGD in latent space. In both cases we use 1000 iterations of SGD, with a learning rate of .01 and no momentum. The loss is an L1 pixel reconstruction loss, resulting in the final images displayed in figures 1 and 2 of the main paper.

---

[1]https://pytorch.org/hub/facebookresearch_pytorch-gan-zoo_pgan

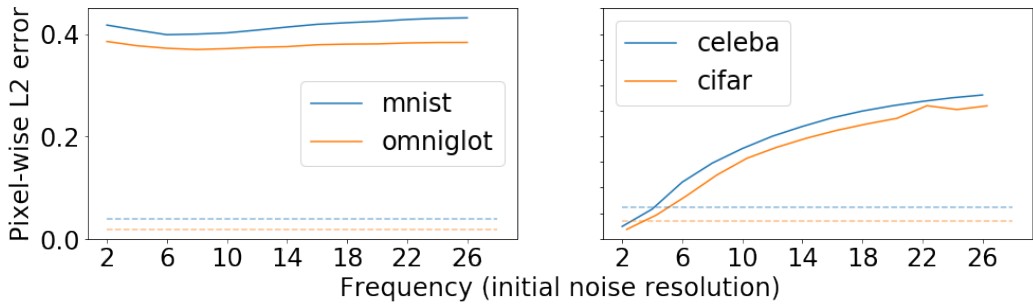

Figure 11: L2 pixel reconstruction error as a function of noise frequency, $[0, 1]$-scaled. Dotted lines represent reconstruction loss on novel domain images (EMNIST for the MNIST model, CIFAR-100 for the CIFAR-10 model, etc.

## A.3 QUANTITATIVE EVALUATION

Fréchet Inception Distance (FID) (Heusel et al., 2017) compares the distributions of embeddings of real ($p_r(x)$) and generated ($p_g(x)$) images. Both these distributions are modeled as multi-dimensional Gaussians parameterized by their respective mean and covariance. The distance measure is defined between the two Gaussian distributions as:

$$d^2((\mathbf{m}_r, \mathbf{C}_r), (\mathbf{m}_g, \mathbf{C}_g)) = \|\mathbf{m}_r - \mathbf{m}_g\|^2 + Tr(\mathbf{C}_r + \mathbf{C}_g - 2(\mathbf{C}_r \mathbf{C}_g)^{\frac{1}{2}}) \tag{7}$$

where $(\mathbf{m}_r, \mathbf{C}_r)$ and $(\mathbf{m}_g, \mathbf{C}_g)$ denote the mean and covariance of real and generated image distributions respectively (Shmelkov et al., 2018). We use a port of the official implementation of FID to PyTorch[2]. The default pool3 layer of the Inception-v3 network (Szegedy et al., 2016) is used to extract activations. We use 5k generated images to compute the FID scores, and sample 3 uniformly distributed points along each sampled interpolation path. To ensure domain fidelity of generated images, we use a binary classifier to distinguish whether the images come from the train distribution or the test set. Classifiers have the same architecture as other experiments, as described in Section A.1.

## A.4 NOISE RECONSTRUCTION

We analyze the ability of a trained AE to reconstruct uniform pixel noise of different frequencies. We simulate noise frequency by sampling noise at small resolutions and scaling up the resulting map to the desired image resolution ($28 \times 28$ for handwritten characters, $64 \times 64$ or $128 \times 128$ for 3-channel color images). Fig. 11 plots the ability of trained AEs to reproduce noise patterns over given frequencies, averaged over 1000 trials. Networks have a clear low-frequency bias - though interestingly, the handwritten character datasets reach their minimum reconstruction error at a frequency level of 6-8, a possible manifestation of a learned bias toward penstroke thicknesses or certain-sized pockets of negative space associated with handwritten characters. Most tellingly, reconstruction error for novel images (dotted lines) is significantly lower than for noise of any frequency for handwritten character models, and most frequencies for natural image models. This suggests clearly that the network has learned a particular image distribution that is not reflected by uniform noise - an image prior.

It is also worth noting in what ways the AEs fail when reconstructing noise. Figs 12 and 13 show reconstruction attempts for random sampled noise at the given frequencies. It is clear that the hand-written character models struggle to abandon a strong, learned penstroke prior. Natural image networks are better at encoding noise, but also demonstrate a clear failure mode at high frequencies where they extract low-contrast, lower-frequency patterns and ignore the higher-frequency input. The Celeb-A model attempts to compensate for this by adding high-frequency ripple artifacts, visible also at some lower frequencies, probably reflecting a learned hair prior.

---

[2]https://github.com/mseitzer/pytorch-fid

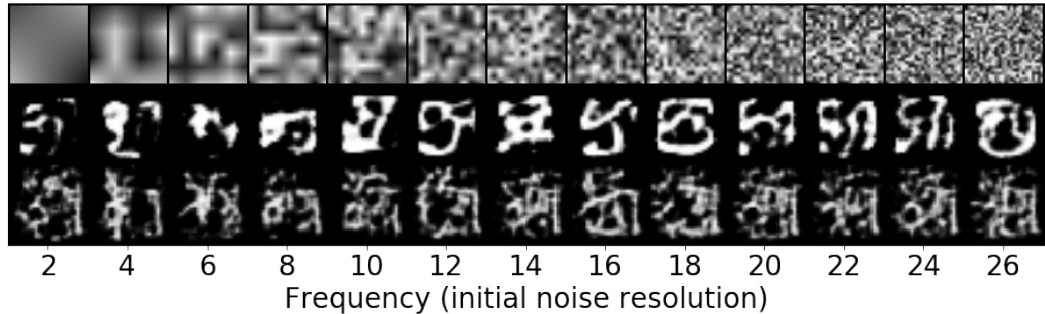

Figure 12: Attempted reconstructions of single-channel uniform noise. First row is noise, second is the MNIST model, third is the Omniglot model. Both models clearly struggle to abandon a learned penstroke prior.

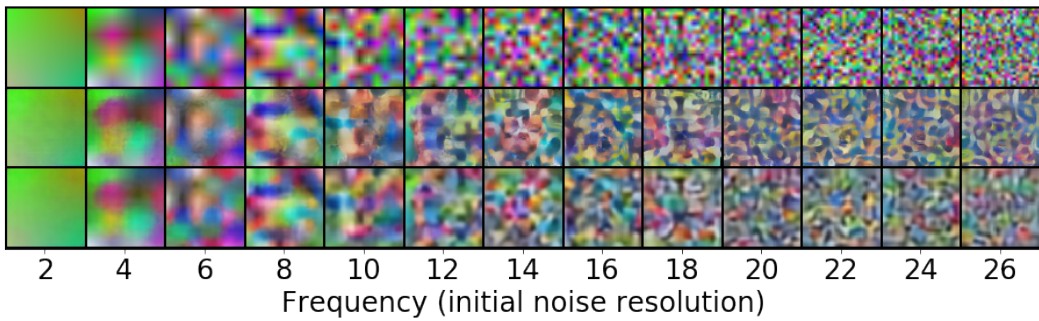

Figure 13: Attempted reconstructions of three-channel uniform noise. First row is noise, second is the Celeb-A model, third is the CIFAR-10 model. As noise frequency climbs, the networks progressively abandon the input signal and attempt to extract a lower-frequency pattern.

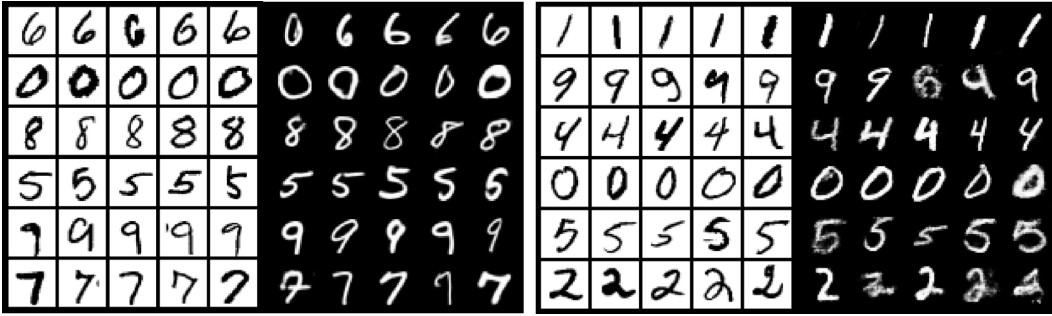

Figure 14: Black digits on white background are seed images, and white digits on black background are novel synthesized images. Neural statistician (left) and DAGAN (right) produce the desired behavior on training classes. Thus failures on novel classes clearly represent a failure to generalize.

## A.5 FEW-SHOT GENERATION BASELINES

We re-implement DAGAN using the architectures from our other experiments, with hyperparameters the same as for WGAN-GP training. The sole difference is that the critic network now takes in a pair of images, so the number of input channels is 2 instead of 1. We implement the critic network as a WGAN-GP. Fig. 14 shows that the network converges nicely on the training data: it successfully produces novel images from the same class as the initial seed images.

Neural statistician uses a much more complex network architecture, and involves many additional hyperparameters, making direct re-implementation difficult. Instead we run the publicly available code[3] as-is, keeping all hyperparameters the same. We run the omniglot experiment, and simply replace the omniglot training dataset with MNIST. Fig. 14 shows that like DAGAN, the network converges nicely and produces the desired behavior on the training data. The failure of these approaches on EMNIST is thus truly a failure of generalization.

## A.6 ADDITIONAL ILLUSTRATIVE EXAMPLES

Selected sets of interpolated image pairs from each of our four dataset regimes, demonstrating that AugIntAE performs smooth and intuitive interpolations where AEs and IntAEs produce artifacts. Images are organized as in the paper, with rows in each cell correspond to pixel fade, AE, IntAE, and AugIntAE. These are followed by four sets of randomly chosen pairs illustrating average-case performance, again one for each of our four dataset regimes. Begins on following page.

---

[3]https://github.com/conormdurkan/neural-statistician

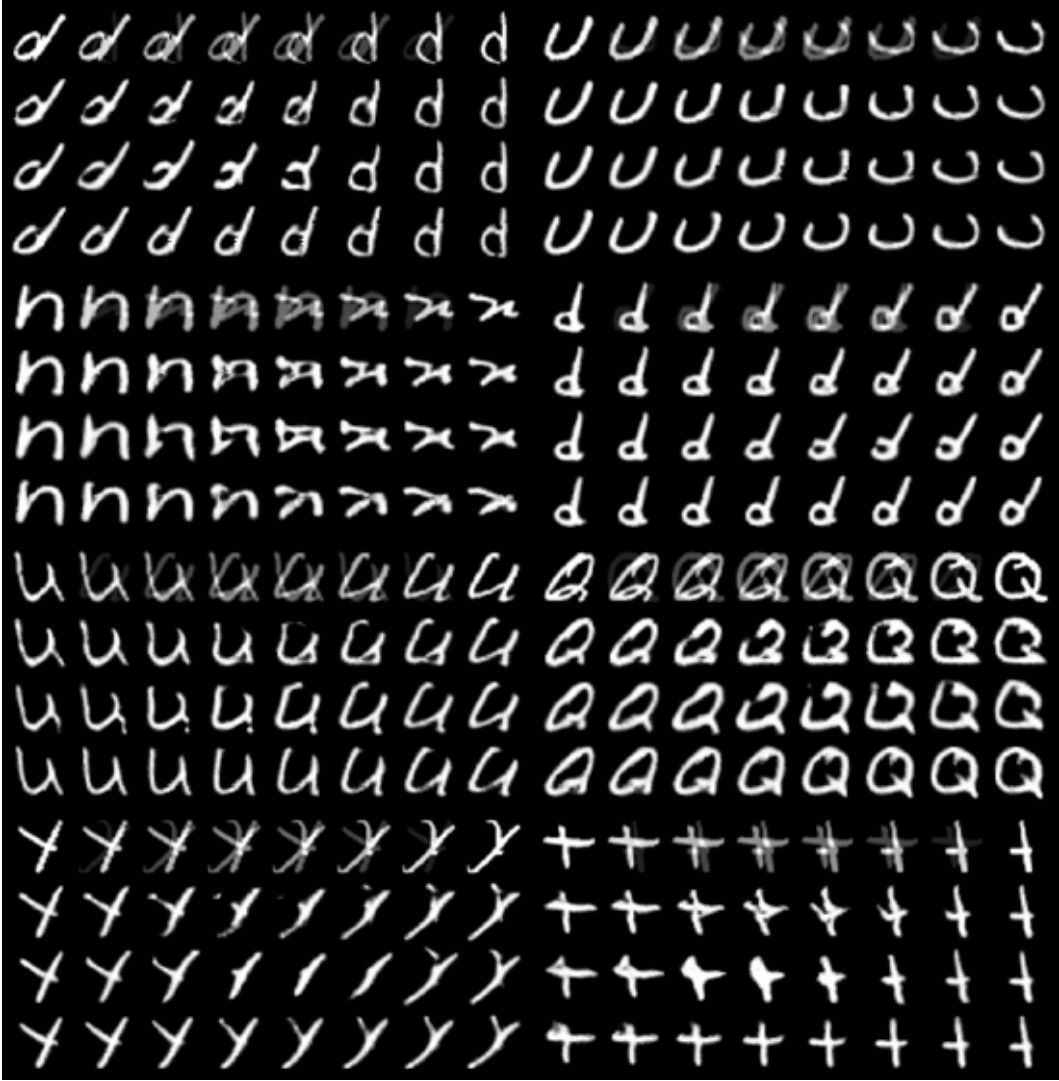

Figure 15: **Illustrative EMNIST pairs**. Each image pair contains artifacts in the AE/IntAE models that AugIntAE is able to avoid.

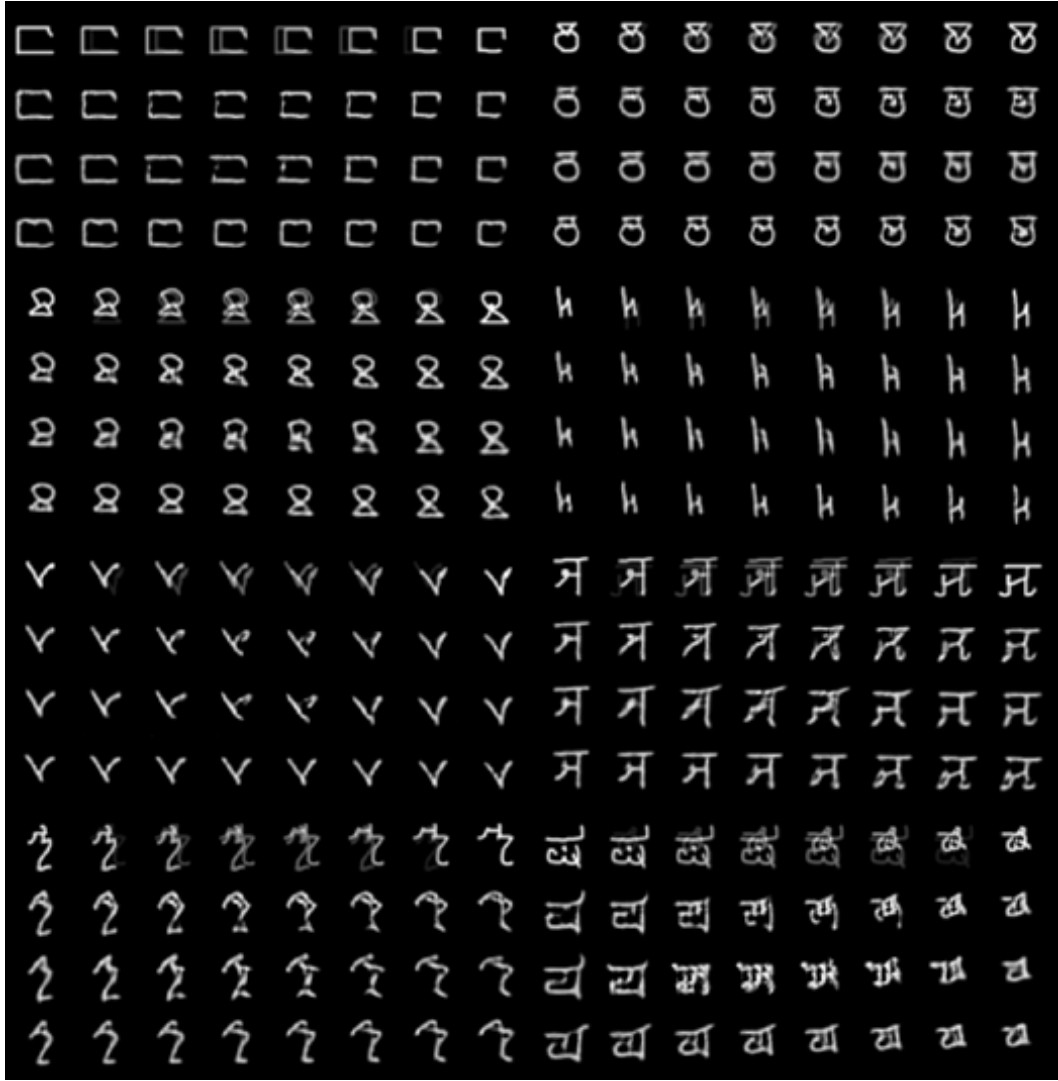

Figure 16: **Illustrative Omniglot pairs**. Each image pair contains artifacts in the AE/IntAE models that AugIntAE is able to avoid.

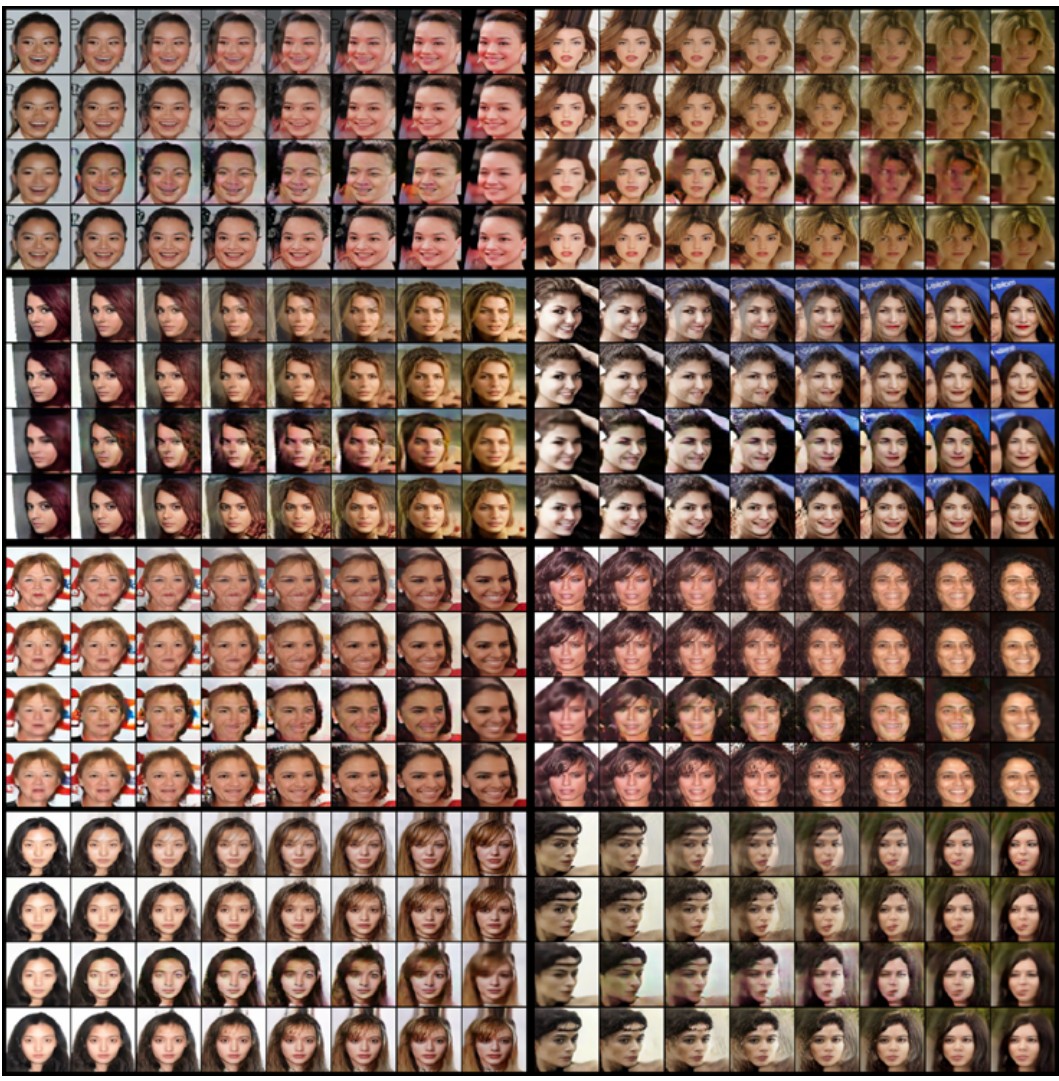

Figure 17: **Illustrative Celeb-A pairs**. Each image pair contains artifacts in the AE/IntAE models that AugIntAE is able to avoid. AEs generally produce transparency and silhouette artifacts around the hairline, mouth, and chin, while IntAEs produce nonsmooth interpolations, unrealistic head contours, and/or color artifacts. We note that AugIntAE is prone to smoothing away fine details, sometimes to a greater degree than AE.

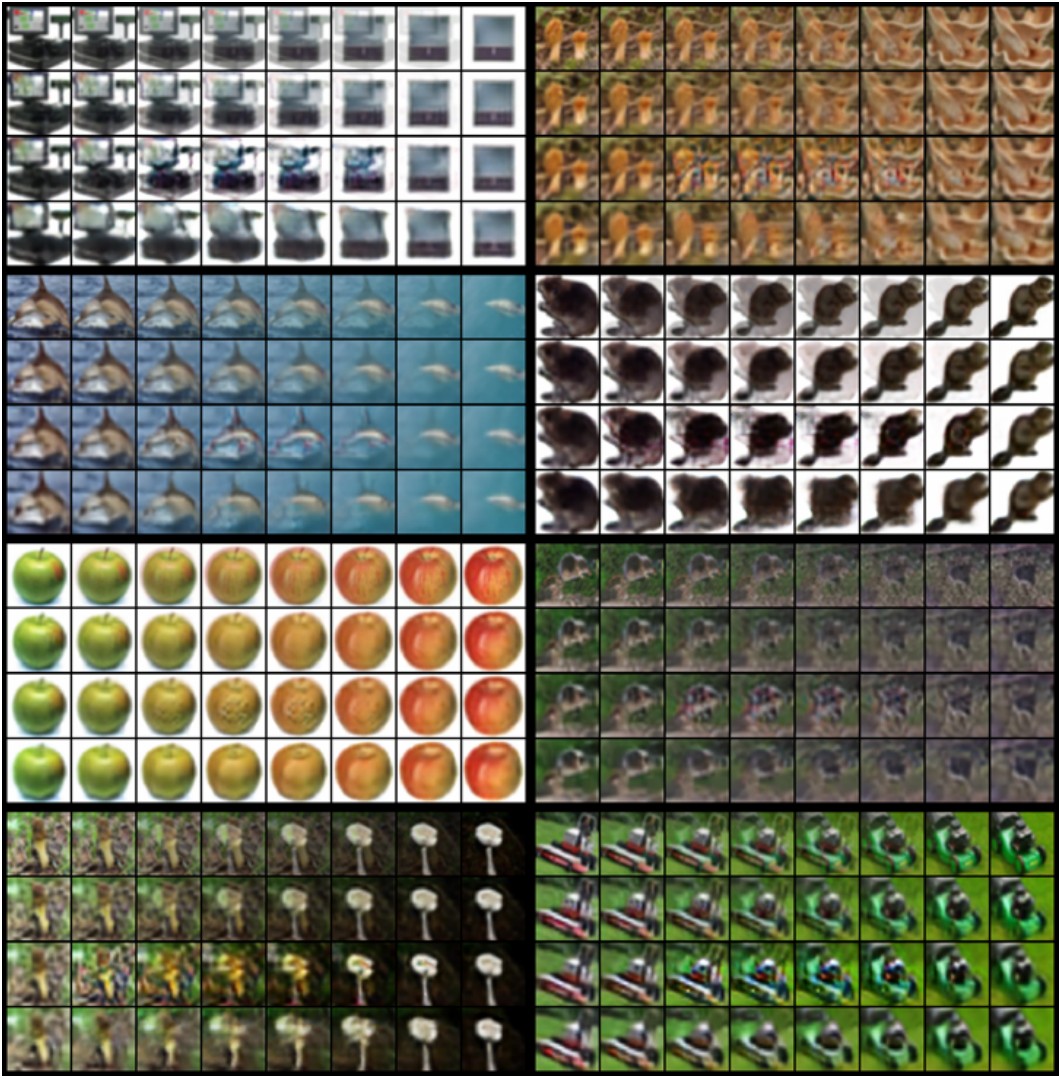

Figure 18: **Illustrative CIFAR-100 pairs**. AEs are largely indistinguishable from pixel fade, while IntAEs produce color and texture artifacts. AugIntAE does not always successfully preserve semantic content through the interpolation, but the interpolation at least usually makes sense. Of particular note are the apples on the left-hand side: rather than fade away the stem, the AugIntAE model retracts it into the apple! However, we again note that the more sensible AugIntAE interpolation comes at the cost of smoothing away fine details.

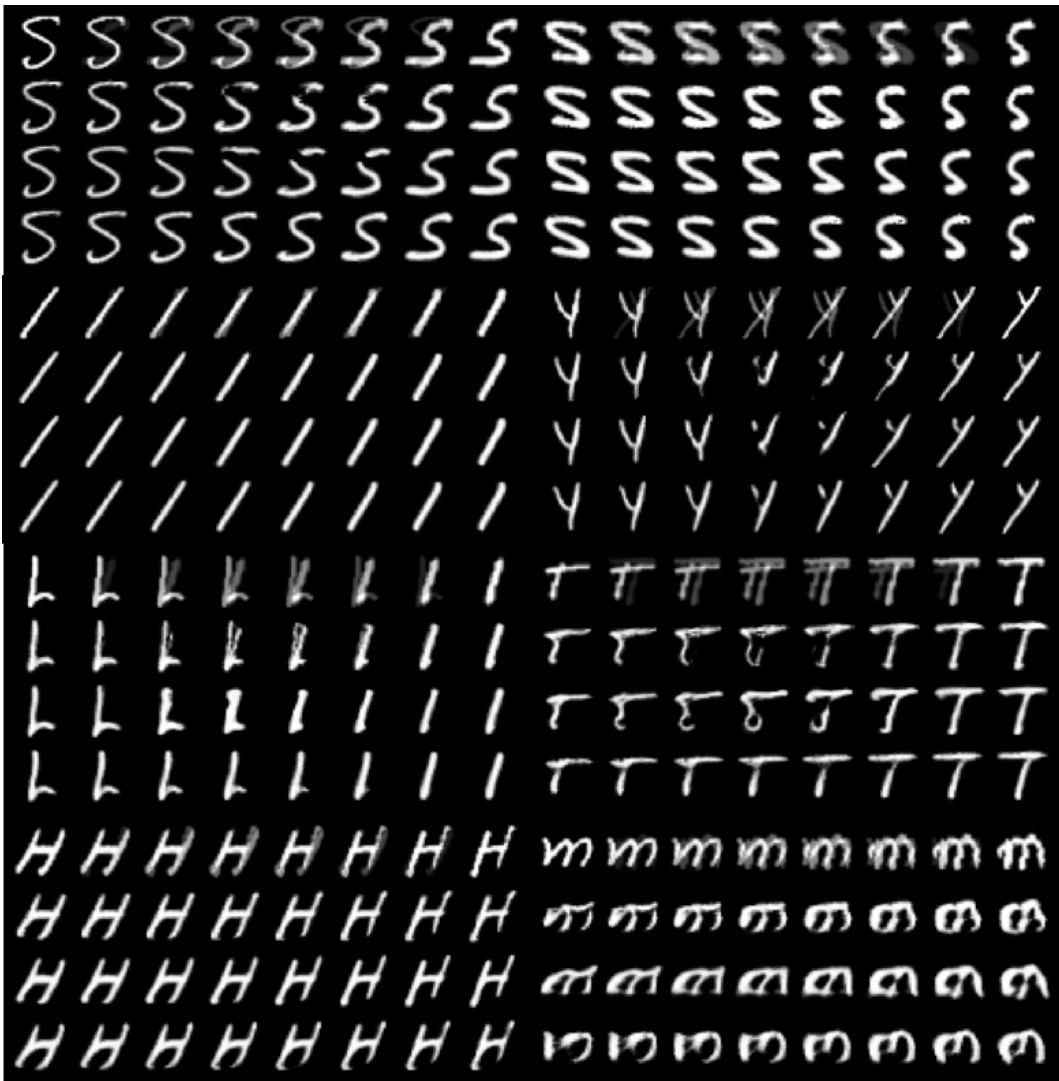

Figure 19: **Random EMNIST pairs**. AugIntAE is superior in some cases, and gives roughly equal performance in the remainder.

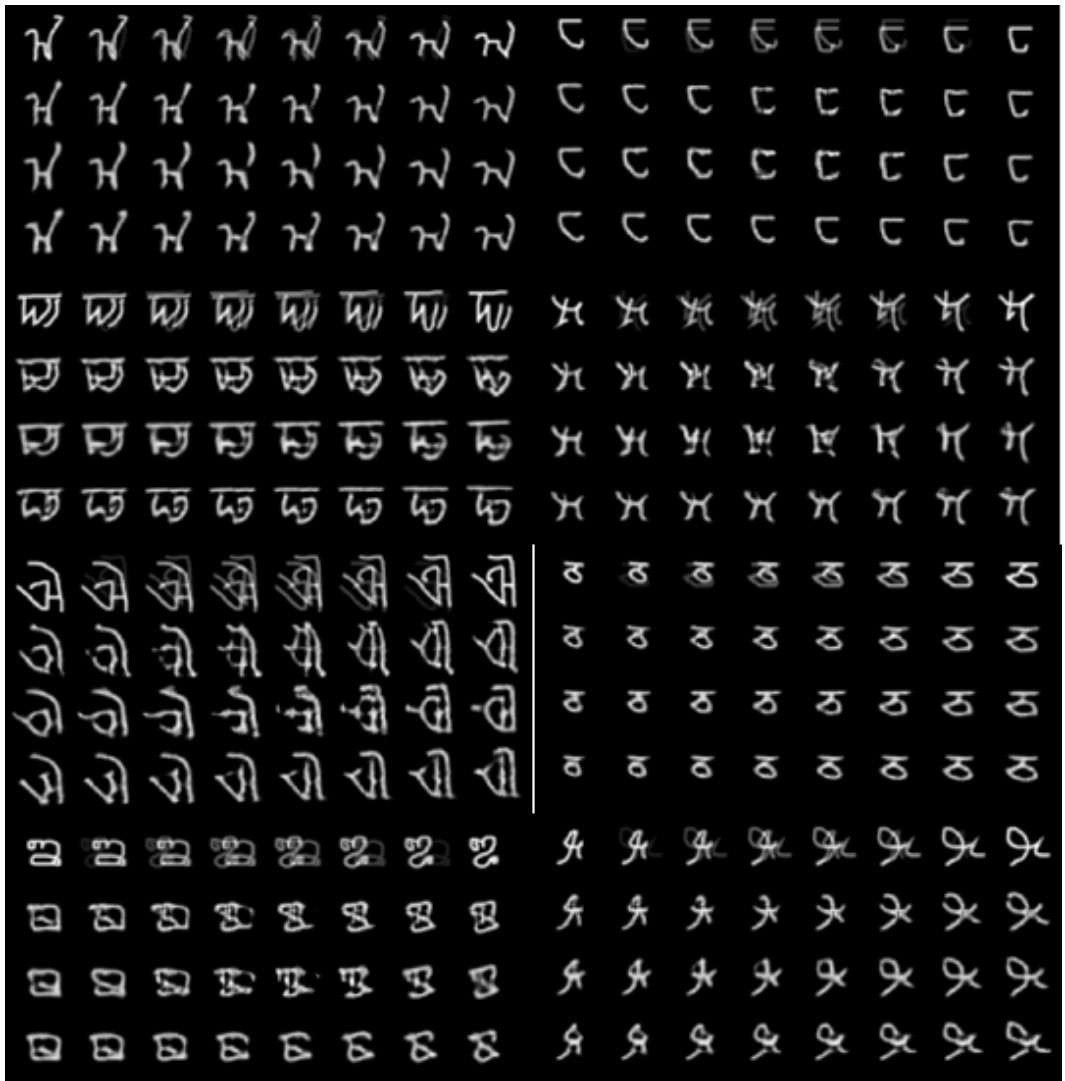

Figure 20: **Random Omniglot pairs**. AugIntAE successfully avoids many visual artifacts, but also tends to oversimplify complex shapes.

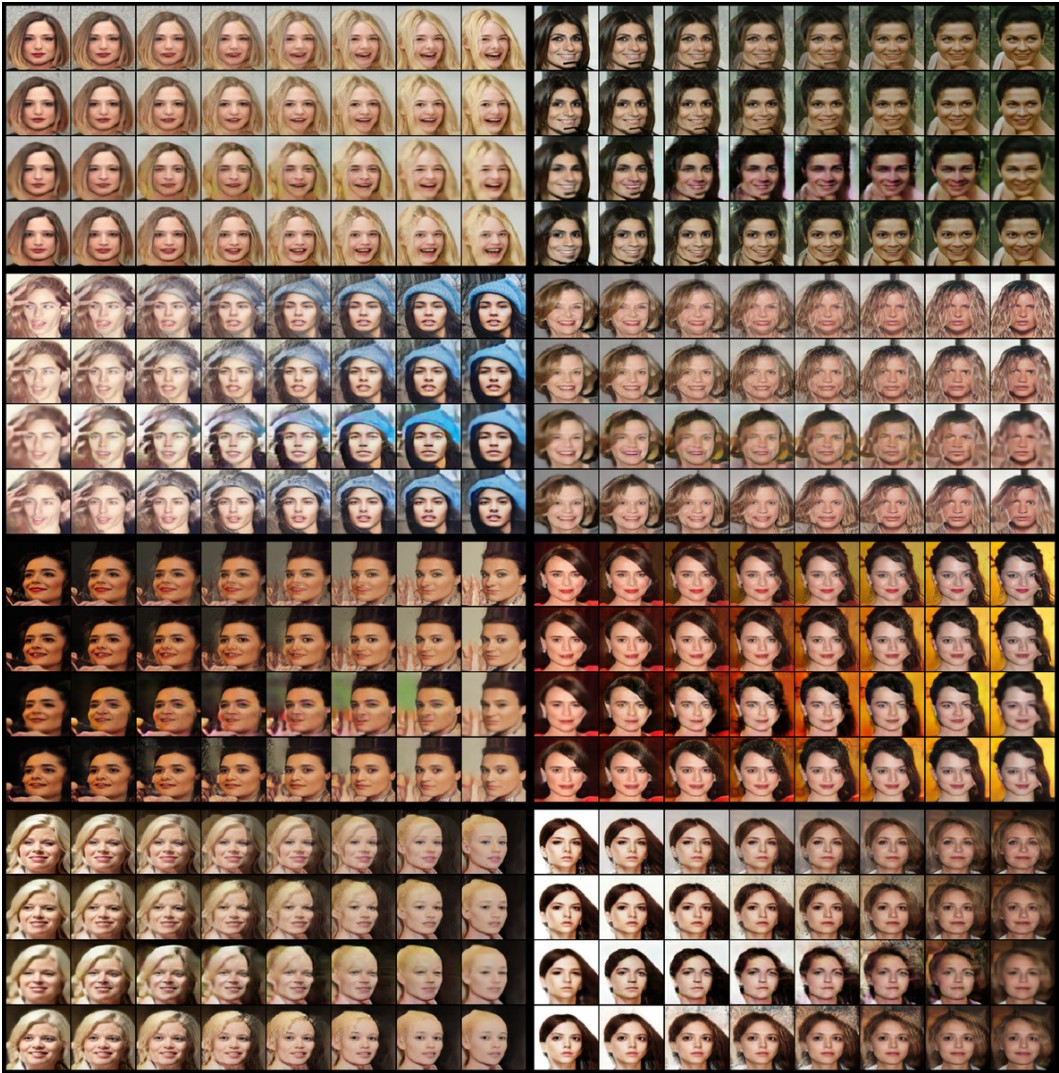

Figure 21: **Random Celeb-A pairs**. Pixel fade is actually a reasonably strong baseline on Celeb-A because of how the images are aligned, so in many cases the models all arrive at the same reasonably good interpolation.

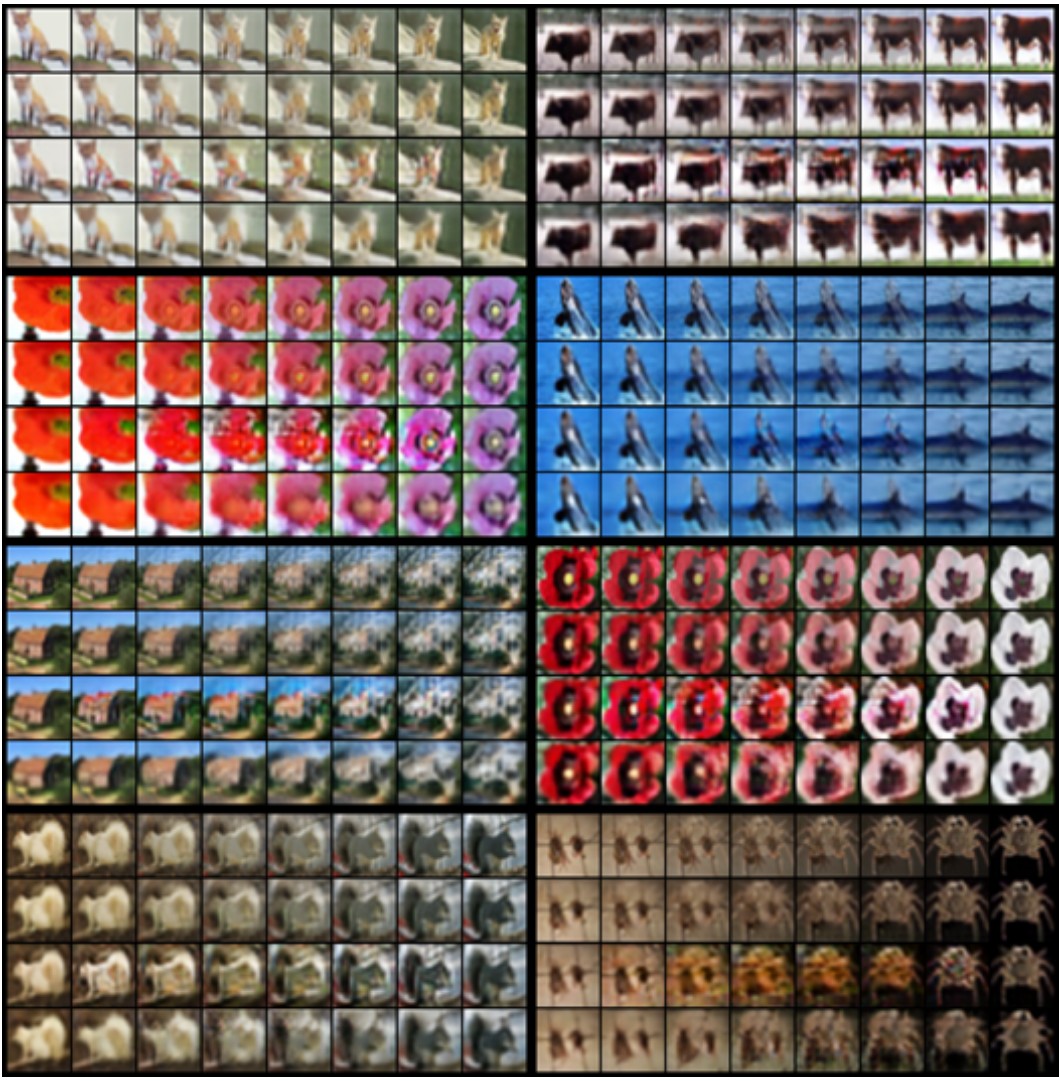

Figure 22: **Random CIFAR-100 pairs**. There is clear room for future work - in some cases the AugIntAE output is unacceptably blurry, and the AugIntAE interpolation frequently fails to preserve semantic content.

