# OpenReview forum: "Augmentation-Interpolative AutoEncoders for Unsupervised Few-Shot Image Generation"
_ICLR.cc/2021/Conference — Reject_

### Official Review · AnonReviewer4 · 2020-10-27
**Augmentation-Interpolative AutoEncoders for Unsupervised Few-Shot Image Generation**

**Rating:** 4
**Confidence:** 4

**Review:**

#######################################

Summary:

The paper investigates the generative model which generalizes to new domain with limited samples.  Authors firstly explore the current hot generative models:  VAEs and GANs, and experimentally find that both VAEs and GANs fail to learn a model which generalizes  well to novel domain.  Interestingly, AutoEncoders exhibits effective performance of the generalizability to new domain. With the encouraging insight, authors further approach Augmentation-Interpolative AutoEncoders. Specially,  the paper firstly augments the training sample to get the  input pair, and extracts the latent feature by the sharing Encoder. A weighted sum of both features is conducted to form the mixed feature, which further is taken as input for the decoder to synthesize the output sample. Differently   the paper performs the reconstruction loss between the output and the mixed input which sum the input pair with the same from to the one of the latent space.

#######################################

Pros:

+To my background, mixing the latent space from the  input pair is new.

+Unlike most papers which explore VAES and GANs, this paper focuses on AEs, and experimentally find its generalizability to new domain.

+This paper is well-written, and easy to follow. The organization is clear, which firstly introduces the drawback of current hot generative models, then indicates the insight that AEs is able to generalize well to novel domain, and finally propose augmentation-Interpolative AutoEncoders.

+The framework is clear to introduce the proposed method.

+Authors show quantitative and qualitative results to support the proposed method which generalizes well to new domain.

#######################################

Cons:

-The idea is little simple, and lack of enough novel. To my best knowledge, the data augmentation is able to increase the generalizability of model, and reduce the domain gap between the train and the test. In this paper, authors indicate AEs has good generalizability, which is interesting, but the proposed method is simple. In fact, for complex dataset, the visualized result is not convincing.

-I am thinking the proposed techniques push the decoder remember the sum of input features.   The same form of both mixing  feature and input pair makes the decoder just remember which feature is from the input and which one form the augmented input.

-Figure 1 is not convincing. Current paper[1] has shown great result on styleGAN, even target domain is from far domaon.  why authors utilize PGAN?

-In this paper, Omniglot train and test are used to evaluate the generalizability. I am wondering why authors consider train and test which extremely close. Please collect me if I am wrong.

-In part of experiment, authors leverage the baseline (Neural Statistician and DAGAN ) which is old. Are there latest paper?


[1] Image2StyleGAN: How to Embed Images Into the StyleGAN Latent Space? ICCV2019

---

> ### Author Response · Authors · 2020-11-17
> **Point by point responses for Reviewer 4:**
>
> Thank you for your comments and feedback! We address concerns in order:
> - First, we believe our contribution is different from the perception given in this review. Our proposed approach (AugIntAE) is not merely data augmentation; instead, it is a novel way of training generative models to interpolate. Second, we agree that our proposed approach is simple: we (and Reviewer 2) view this as a feature, not a bug. As reviewers have no doubt seen, our field has a veritable graveyard of complex, uninterpretable models that yield minor performance improvements; if we find a simple, easily understood insight with a consistent and disproportionate impact, this is to be celebrated. Third, note that our results show that our model interpolates successfully in completely unseen domains. Generative quality on seen domains, which reviewers in this field are probably more used to, will of course be higher. The fact that we can get non-trivial generation to unseen domains at all is in itself a big result.
> - The proposed trivial solution of memorizing the training set does not minimize the interpolative training objective. The recovered image is a midpoint of two different sampled augmentations, not the original unaugmented image. As such, any augmentation of the image could conceivably be sampled at some point during training as the target reconstruction. In any case, our generalization results to entirely new domains should indicate that the network is not simply memorizing the training images.
> - Figure 1 is intended only as an illustrative example. [1] does reproduce images from a novel domain. However, they do so by optimizing in an 18*512=9216 dimensional latent space, which is sufficient to recover the image even from a random, untrained network ([1] Figure 5, row e). While certainly useful practically, this does not disprove our claim that a trained GAN does not generalize to new classes by default (see for example [1] Figure 5 rows c and d). We chose PGAN more or less arbitrarily as a generic representative for large pretrained GANs.
> - Omniglot train and test are not similar; they contain disjoint classes from different alphabets. For example, the train set may include letters from English, while the test set may include characters from Arabic. The difference between Omniglot train and Omniglot test is roughly comparable to the difference between letters and digits.
> - There is very little work that attempts to generalize image generation to new domains. Neural statistician and DAGAN were the two latest baselines whose results we were able to reproduce (alongside IntAE). Other than the approaches that fall under the IntAE umbrella, we are not aware of any more recent advances in few-shot image generation.

---

### Official Review · AnonReviewer1 · 2020-10-28
**Image generation for dataset with few examples**

**Rating:** 4
**Confidence:** 5

**Review:**

PROS:

1. The work is well motivated. The topic about doing augmentation for few shot learning has potential value in real application.
2. The authors made interesting discovery on the generalization of AE.
3. The authors also made a good point in utilizing AE to do augmentation for few shot learning

CONS:

The contributions of this work are a bit weak and not enough for ICLR. The work needs to be polished more seriously and it is not ready for publishing. The assumption/settings about the method is not clear/correct, some statements are too strong, the implementation is not clear, the settings for the experiments are omitted. Detailed concerns are listed as follows.

1. What’s the relationship between the large available X and the few shot X’? Given a few shot X’, will it work if we use an arbitrarily selected large X? if no, dose there have any constrain for choosing a proper X, should be seriously analyzed.
2. The definition of a models’ generalization in this paper seems in contrary to the existing literatures (where the VAE are claimed to generalize better than AE), a discussion on this topic is important in understanding the contribution of this work.
3. How to obtain the reconstruction for VAE method in Figure2? Since there is a sampling process in obtaining the latent code, is the latent code obtained via z=mu(x)+sigma(x)*eps, eps~N(0,I)? If yes, what about the reconstruction with z=mu(x)?
4. I am curious about the range of the value for latent code learned by AE, also what's the difference of the latent space between AE and AugIntAE?
5. The WGAN-GP on CIFAR100 as shown in Table 2 seems not trained well, by referring to [1], the FID can be as small as 15.6, however the reported performance here is 54.3, which is much higher than that in previous literature.  In section 5, the authors used a shallow network for illustration, which makes it difficult for the readers to evaluate the reliability of the experiments, it is easier if an existing model is employed, i.e. WGAN-GP.
6. The training process for the method is very cryptic, since two datasets are mentioned in this work (one in large scale and another with few shot), which one is the case used in this work, trained jointly on these two datasets, pretrained on the large scale one and transfer to the few shot case, or only trained on the large scale dataset?
7. How many images are used for the few shot dataset?
8. I doubt the statement on few-shot generation is correct, once the AugIntAE is trained on some dataset, can we really apply it to “any” set of seeds (images)? This statement is really strong and the experiments didn’t support this statement properly, i.e. the model is trained  and tested on relatively related datasets.
9. The classification error in Table 2 is confusing, e.g. how to obtain this value?
10. Table 3 is not referred.
11. About Table 3. How many real images (except the augmentations) are used to train the classifier? Since the augmentation is obtained from two randomly selected images, how to define the label for this augmentation? e.g. what is the label for the augmentation obtained from 3 and 4?
12. A quantitative evaluation for the generated images in experiments of Fig.8 and Fig.9 is necessary, e.g. FID or Kernel Maximum Mean Discrepancy (KMMD). What’s the performance when compared to some recent GAN based method[2]? what’s the pros and cons by comparison?


[1] Shmelkov K, Schmid C, Alahari K. How good is my GAN?[C]//Proceedings of the European Conference on Computer Vision (ECCV). 2018: 213-229.
[2] Zhao S, Liu Z, Lin J, et al. Differentiable augmentation for data-efficient gan training[J]. arXiv preprint arXiv:2006.10738, 2020.

---

> ### Author Response · Authors · 2020-11-17
> **Point by point responses for Reviewer 1:**
>
> Thank you for your comments and feedback! We address concerns in order:
> 1. It is almost certainly the case that a semantic relationship of some sort must exist between X and X’. However, understanding how to choose a “proper” dataset X remains an open area of research. The fields of few-shot learning and transfer learning have yet to solve or even propose a rigorous solution to this problem for classifiers. We believe that even demonstrating that image generators can be made to generalize is a valuable contribution in and of itself.
> 2. In this work “generalization” refers to generalizing to new domains, not to generalizing to new images from the same domain. We are not aware of claims that VAEs generalize better than AEs in this sense (and have cited work suggesting that they don’t). We have emphasized this distinction in the paper.
> 3. We use z=mu(x), no sampling is involved. Added clarification.
> 4. Our models send all latent representations to the unit ball, but there is no other prior, so direct comparison is difficult. We find that AugIntAE uses a smaller portion of the spherical surface (feature values range from ~-.4 to .4, as opposed to -.6 to .6 for AE) but this could just indicate that the latent space exists at a higher resolution only with less curvature. Our results indicate that the AugIntAE latent space is significantly smoother than those from prior methods, but beyond that we have found no significant differences.
> 5. The FID of 54.3 comes from a CIFAR10 model evaluated on CIFAR100, so a bad score is expected. Our label-free CIFAR100 WGAN-GP scored on CIFAR100 (the standard evaluation setup) gets 42.0, which is comparable to the class-conditioned DCGAN (whose neural architecture we share) given in [1].
> 6. We train only on the larger dataset. Our main finding is that our generator can interpolate out-of-the-box on the smaller dataset without any adaptation. We have added clarification in the paper.
> 7. Each “few-shot” run that we evaluate or visualize uses 2-6 images. For the quantitative scores in Table 2, interpolations are synthesized from pairs of images drawn randomly from the testing dataset.
> 8. You’re correct, this is overstated. We meant only that the method can sample novel images given a set of seeds and no other information. We did not mean to imply that the method works automatically for *any* arbitrary seed set (see point 1). We have clarified the language in the paper.
> 9. We have added a subsection in results that explains this better. A separate classifier is trained to distinguish train from test images. If the image generator generalizes, then images synthesized from test seeds should be classified as test images. We report the predicted rate of test-set membership.
> 10. Thank you, added the reference
> 11. We use the same set of all the labeled images to train the classifier. Because we have class labels in this setting, we interpolate only between images of the same class, and preserve the label. We have clarified this in the paper.
> 12. FID and classification scores for Fig.8 and 9 images can be found in table 2. We view GAN-adaptation and sparse-data methods as largely orthogonal to our work: our model can be used as a pretrained initialization for these methods, incorporating domain knowledge from a larger dataset. Added [2] to related work.

---

### Official Review · AnonReviewer2 · 2020-10-28
**Interesting and simple (which is positive) method - but experiments are lacking**

**Rating:** 5
**Confidence:** 4

**Review:**

The general idea of the paper is interesting: when using an AE one can use the constraint that "interpolated images" should also correspond to "interpolated latent codes". While the idea is interesting, the experimental results are not really that compelling.

The proposed approach in section 4 is an interesting and well motivated extension to Interpolative AEs. The idea is quite simple and the proposed setting to use synthesized and augmented images for the above mentioned interpolation constraint seems interesting to explore.

The main weakness of the paper are the experimental results in my view.

While qualitatively (and potentially hand-picked) examples seem to show that the proposed approach is working well, the experiments are not sufficient to convince me as a reviewer about the power of the approach. Let me be more specific

- In general, quantitative results are rare and thus it is close to impossible to assess the performance of the proposed method. In essence mostly qualitative results are shown that are obviously anecdotal only. An exception is table 2, where FID and an error is shown. While I understand the FID score, I am lacking comparisons to FID scores for these models trained to the "same" domain. Otherwise it is unclear how good the numbers really are. Also, the error numbers where not entirely clear to me what they correspond to.

- An important ingredient and component of the approach is the way the synthesized images are obtained via augmentation. While the reader get a vague idea about what kind of augmentation is used, there is no experiment that shows which kind of augmentation is necessary and which kind of augmentation will break the system. In fact, without such an "ablation-type" experiment the paper is not particularly insightful. To me some experiments around this essential component the paper is incomplete and should not be accepted.

- Finally, somewhat linked to the previous comment, the paper does not really show failure modes (with the somewhat too obvious failure mode given in fig 3 right) to understand the limitations of the proposed method


So overall the proposed method is interesting and simple (which is positive) - but the experimental results are not convincing and complete enough to justify acceptance at ICLR.

Update after the rebuttal:

Thanks for the responses. Given that the other reviewers also raise serious issues I will stick with my initial rating. The paper seems not to be ready for publication at ICLR

---

> ### Author Response · Authors · 2020-11-17
> **Response to Reviewer 2: clarified quantitative results, added ablations**
>
> Thank you for your comments and feedback! We agree that qualitative evaluation is difficult and thus quantitative scores are very important. We have added a new subsection devoted to discussing/explaining the quantitative results and have expanded and clarified table 2. We believe that the clarified language/tables address your concerns, in particular:
> - FID scores for models trained in the same domain are in fact included in table 2. The right-hand column "WGAN_GP (test)" contains scores for a generative model trained and evaluated on the same test dataset. Our models approach but do not exceed these oracle scores.
> - We have added an ablation study on the MNIST/EMNIST setting. We find that no one augmentation makes or breaks the model - they act synergistically, hence the kitchen sink approach adopted in the remaining experiments.
> - The additional results in supplementary present non-handpicked results and discuss failure modes, eg. figs 17-21. We find that the network tends to simplify complex shapes (omniglot), blur out small details (celeba/cifar), and completely fail when semantic interpolation is inherently difficult (cifar).

---

### Decision · Program_Chairs · 2021-01-07
**Final Decision**

**Decision:**

Reject

**Comment:**

This meta-review is written after considering the reviews, the authors’ responses, the discussion, and the paper itself.

The paper proposes a training scheme for autoencoders, involving data augmentation and interpolation, that results in autoencoders for which interpolations in the latent space lead to meaningful interpolations in the image space. The paper notices that this property carries over reasonably well to datasets different from the training one.

The reviewers point out that the idea is interesting (R1, R2, R4) and simple (R2), but the experiments are substandard (R2, R4) and presentation is at times suboptimal (R1). Overall consensus is towards rejection. Authors addressed some of the concerns in their responses, but failed to convince the reviewers to change their evaluations.

I agree with the reviewers and recommend rejection at this point. The idea is indeed interesting and could be publishable if presented and evaluated well, but in the current manuscript the presentation is at times unclear or somewhat misleading (e.g. presenting the method as a general image generation method, not an interpolation method) and the experiments are reasonable, but not quite convincing, mainly because the architectures and the baselines are outdated (as also pointed out by R1 and R4). I encourage the authors to further improve the paper and resubmit to a different venue.